# Cross-sectional centiles of blood pressure by age and sex: a four-hospital database retrospective observational analysis

David Wong ![ID],[1,2] Stephen Gerry,[3] Farah Shamout,[4] David A Clifton,[4] Marco A F Pimentel,[4] Peter J Watkinson ![ID] [5]

¹Centre for Health Informatics, The University of Manchester, Manchester, UK
²Computer Science, The University of Manchester, Manchester, UK
³Centre for Statistics in Medicine, University of Oxford, Oxford, UK
⁴Institute of Biomedical Engineering, University of Oxford, Oxford, UK
⁵Kadoorie Centre for Critical Care research and Education, Oxford University Hospitals NHS Trust, Oxford, UK

**Correspondence to**
Dr David Wong;
david.wong@manchester.ac.uk

## ABSTRACT

**Objectives** National guidelines for identifying physiological deterioration and sepsis in hospitals depend on thresholds for blood pressure that do not account for age or sex. In populations outside hospital, differences in blood pressure are known to occur with both variables. Whether these differences remain in the hospitalised population is unknown. This database analysis study aims to generate representative centiles to quantify variations in blood pressure by age and sex in hospitalised patients.

**Design** Retrospective cross-sectional observational database analysis.

**Setting** Four near-sea-level hospitals between April 2015 and April 2017

**Participants** 75 342 adult patients who were admitted to the hospitals and had at least one set of documented vital sign observations within 24 hours before discharge were eligible for inclusion. Patients were excluded if they died in hospital, had no vital signs 24 hours prior to discharge, were readmitted within 7 days of discharge, had missing age or sex or had no blood pressure recorded.

**Results** Systolic blood pressure (SBP) for hospitalised patients increases with age for both sexes. Median SBP increases from 122 (CI: 121.1 to 122.1) mm Hg to 132 (CI: 130.9 to 132.2) mm Hg in men, and 114 (CI: 113.1 to 114.4) mm Hg to 135 (CI: 134.5 to 136.2) mm Hg in women, between the ages of 20 and 90 years. Diastolic blood pressure peaked around 50 years for men 76 (CI: 75.5 to 75.9) mm Hg and women 69 (CI: 69.0 to 69.4) mm Hg. The blood pressure criterion for sepsis, systolic <100 mm Hg, was met by 2.3% of younger (20–30 years) men and 3.5% of older men (81–90 years). In comparison, the criterion was met by 9.7% of younger women and 2.6% of older women.

**Conclusion** We have quantified variations in blood pressure by age and sex in hospitalised patients that have implications for recognition of deterioration. Nearly 10% of younger women met the blood pressure criterion for sepsis at hospital discharge.

## INTRODUCTION

Routine measurement of blood pressure is a key component of patient surveillance and diagnosis in hospitals worldwide. Currently, in-hospital assessment of blood pressure is

**Strengths and limitations of this study**

► Changes in blood pressure by age and sex are currently unknown for the hospitalised population.
► A large cross-sectional database of 75 342 patients were used to derive blood pressure centiles.
► Results have implications on how sepsis and other in-hospital deterioration are detected in routine care.
► Though patterns match those seen in out-of-hospital longitudinal studies, cross-sectional analysis can be affected by survival bias and birth cohort effects.

undertaken by comparison to generic population normal ranges.

The ability to use an individual's physiology to monitor them for signs of deterioration may be improved by taking into account factors that affect these normal ranges.[1] For instance, in paediatric medicine, it is accepted that the normal ranges of vital signs vary with age and patients are managed in light of their age-specific normal ranges.[2 3] However, none of the published physiology-based systems for recognising deterioration in hospitalised adults take account of variations in vital signs by age or sex,[4] despite growing evidence that these factors may provide additional information for accurately identifying deterioration.[5 6] For example, the National Early Warning Score 2 (NEWS2)[7] mandates that patients with systolic blood pressure (SBP) below 90 mm Hg require urgent attention and current sepsis guidelines blood pressure criterion are met when SBP is less than 100 mm Hg,[8] both regardless of age or sex.

In populations outside hospital, differences in blood pressure are known to occur with both age and sex.[9] If clinically significant differences also exist in hospitalised adult populations, opportunities for earlier identification and management of patient deterioration may be being missed.

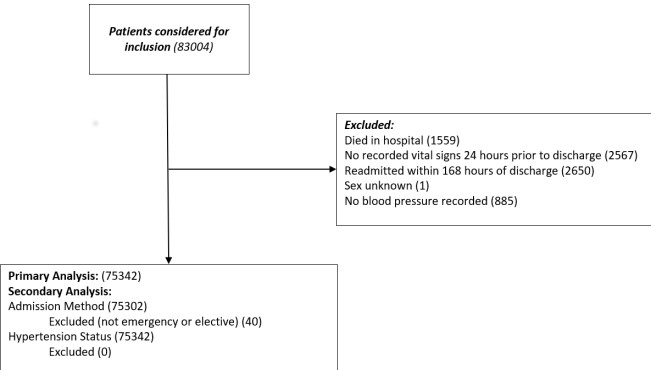

**Figure 1** Consolidated Standards of Reporting Trials diagram showing analysis inclusion criteria.

To quantify these differences, our objective was to define representative centiles of the stable hospitalised population for SBP, diastolic blood pressure (DBP) and pulse pressure (PP) by age and sex via an analysis of a large near-sea-level database of routinely collected vital signs. Description of this group allows inference about unstable patients via one-class classification (novelty detection), which has previously been used when a clinical outcome of interest is relatively uncommon.

## METHODS

We conducted a cross-sectional analysis from a database collated at Oxford University Hospitals (OUH) NHS foundation trust group of hospitals. The OUH consists of four hospitals: an urban teaching hospital, a general district hospital and two specialist hospitals. Our data set included patients admitted to the OUH between April 2015 and April 2017.

All adult patients who were admitted to OUH and had at least one set of documented vital sign observations within 24 hours prior to discharge were eligible for inclusion. Patients were excluded from the analysis if they (1) died in hospital, (2) had no recorded vital signs 24 hours prior to discharge, (3) were readmitted within 7 days of discharge, (4) had missing recordings for age or sex or (5) had no blood pressure recorded.

We collected the final recorded set of blood pressure observations from a patient's first attendance to the OUH hospital group during the study period. This ensured that the centiles were not biased towards repeat attenders or patients with longer hospital stays. Blood pressure was measured using automated devices ratified for clinical use as part of routine clinical care and electronically documented using the SEND e-Obs software.[10] Data were validated for plausible range at the point of entry. Hospital admission time, discharge time, discharge status, ethnicity, admission method and main specialty were also collected for each patient from the hospital electronic patient record (Cerner Millennium, Cerner, Kansas City, Missouri, USA). One investigator (PJW) had access to a small proportion of the database population as part of routine clinical care responsibilities.

Admissions were typed as either elective, emergency or other, according to the admission method code. Codes are defined in full within the NHS data dictionary.[11] The set of ICD-10 codes (I10, I11, I12, I13, I14, I15) was used to determine patients with a primary diagnosis of hypertension.[12]

## Analysis

The characteristics of the study population were described using medians and quartiles for the continuous variables, and frequencies otherwise. We calculated median and representative centiles (1st, 5th, 10th, 25th, 75th, 90th, 95th, 99th) for blood pressure at all ages between 20 and 90, for men and women. SBP and DBP are presented separately. One further measure, the PP was derived as the arithmetic difference between SBP and DBP and was also analysed using the same methods.

In subgroup analyses, we produced separate representative centiles by age and sex for emergency and elective admissions, and for patients without a diagnostic code for hypertension.

Centiles were estimated using the Generalised Additive Models for Location, Scale and Shape framework (GAMLSS).[13] This semi-parametric method provides various options for fitting non-normal distributions to the data. To create smooth centiles across the age range, penalised splines and fractional polynomials were used as smoothing functions. For each vital sign, we assessed six different distributions within the GAMLSS framework: Box-Cox Cole and Green, Box-Cox Power Exponential, Box-Cox-t, Skew Power Exponential type 3, Skew t type 3 and Power Exponential. The best fitting distribution was chosen based on a combination of model fit (Akaike information criterion and Bayesian information criterion)[14 15] and a comparison of fitted versus empirical centiles. Box-Cox t distribution was the best fit for male and female SBP, the Skew t type 3 distribution was chosen for male DBP and male and female PP, and the Skew power exponential distribution was chosen for female DBP. SBP and PP models used penalised-splines as a smoother, while the DBP models used fractional polynomials as a smoother. To ensure fair comparison, the same distribution was chosen for all subgroups within any given vital sign.

All analyses were undertaken using R and the GAMLSS package.[16]

## Sample size

We used all the available data and therefore no formal sample size calculation was undertaken. To ensure that the sample was sufficient, the precision of the centiles was estimated via a bootstrapping procedure, whereby the dataset was sampled with replacement to create a new dataset and the analysis was carried out.[17] This was repeated 50 times. The SD of these bootstrapped estimates was used to calculate the 95% CI for each centile at 2 yearly intervals. Full details of centile values and CIs are provided in online supplementary appendix A.

**Table 1** Characteristics of the study population

| | Female | Male | Total |
|---|---|---|---|
| Total (N=75342) | 39157 (52.0%) | 36185 (48.0%) | 75342 (100.0%) |
| **Patient characteristics** | | | |
| Ethnicity | | | |
| White | 30274 | 26580 | 56854 (75.5%) |
| Mixed | 263 | 261 | 524 (0.7%) |
| Asian or Asian British | 942 | 836 | 1778 (2.4%) |
| Black or Black British | 388 | 363 | 751 (1.0%) |
| Other | 361 | 341 | 702 (0.9%) |
| Not known or stated | 6929 | 7804 | 14733 (19.6%) |
| Age (years) | | | |
| <20 | 1082 | 918 | 2000 (2.7%) |
| 20–29 | 4137 | 3456 | 7593 (10.1%) |
| 30–39 | 4401 | 3391 | 7792 (10.3%) |
| 40–49 | 4995 | 4131 | 9126 (12.1%) |
| 50–59 | 5706 | 5676 | 11382 (15.1%) |
| 60–69 | 5815 | 6538 | 12353 (16.4%) |
| 70–79 | 6081 | 6674 | 12755 (16.9%) |
| 80–89 | 5084 | 4412 | 9496 (12.6%) |
| >89 | 1856 | 989 | 2845 (3.8%) |
| Median age (IQR) | 58 (40–75) | 60 (43–74) | 59 (41–74) |
| **Admission characteristics** | | | |
| Main specialty | | | |
| Medical | 17023 | 13027 | 30050 (39.9%) |
| Surgical | 21202 | 22014 | 43216 (57.4%) |
| Other | 932 | 1144 | 2076 (2.8%) |
| Admission method | | | |
| Emergency | 21542 | 19586 | 41383 (54.9%) |
| Elective | 17323 | 16596 | 33919 (45.0%) |
| Other | 37 | 3 | 40 (0.1%) |
| Hypertension code | | | |
| Yes | 9622 | 10047 | 19669 (26.1%) |
| No | 29535 | 26138 | 55673 (73.9%) |

**Patient and public involvement**

Patients or the public were not involved in the design, or conduct, or reporting, or dissemination plans of our research

**RESULTS**

A total of 83004 patients were admitted to the hospital trust during the period of study and received at least one observation on the SEND e-Obs system. Of these, 75342 patients were included in the study. Blood pressure was missing in 885 (1.17%) records. Other reasons for exclusion are shown in figure 1. Patient and hospital descriptors are shown in table 1.

**Blood pressure centiles**

Centiles by age for SBP, DBP and PP are shown in figure 2 for each sex. Figure 2A shows a progressive increase in median SBP from 122 (CI: 121.1 to 122.1) mm Hg to 132 (CI: 130.9 to 132.2) mm Hg for men between the ages of 20 and 90 years. Younger women had a lower median SBP than younger men (114 (CI: 113.1 to 114.4) mm Hg at age 20 years). By the age of 90 years, median SBP was higher for women than for men (135 (CI: 134.5 to 136.2) mm Hg).

Figure 2B shows that median male DBP peaked at age 50 year (76 (CI: 75.5 to 75.9) mm Hg) with lower median DBP at age 20 years (66 (CI: 65.0 to 66.0) mm Hg) and age 90 years (68 (CI: 67.9 to 68.4) mm Hg). In the female

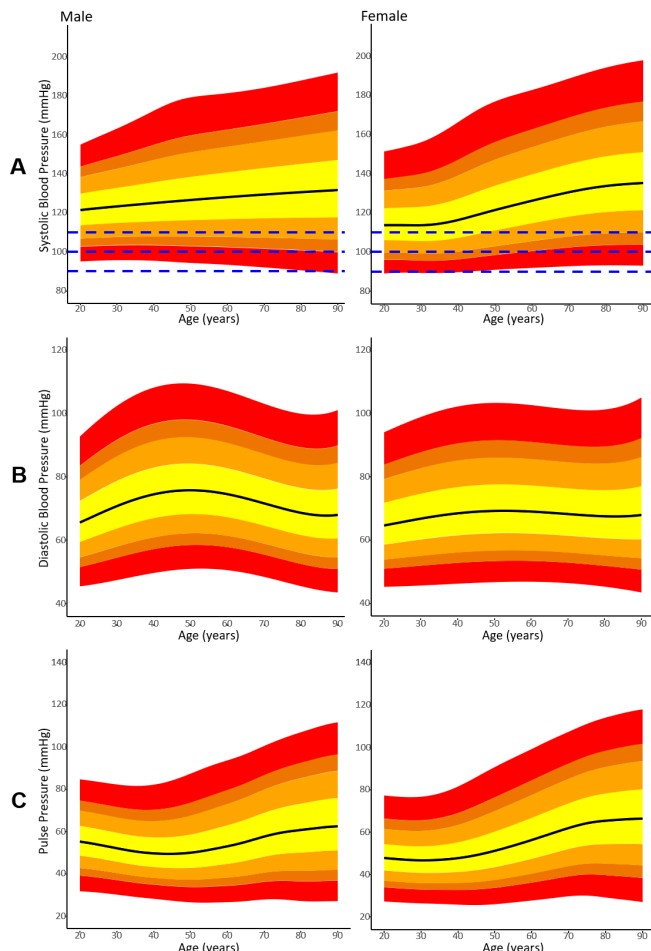

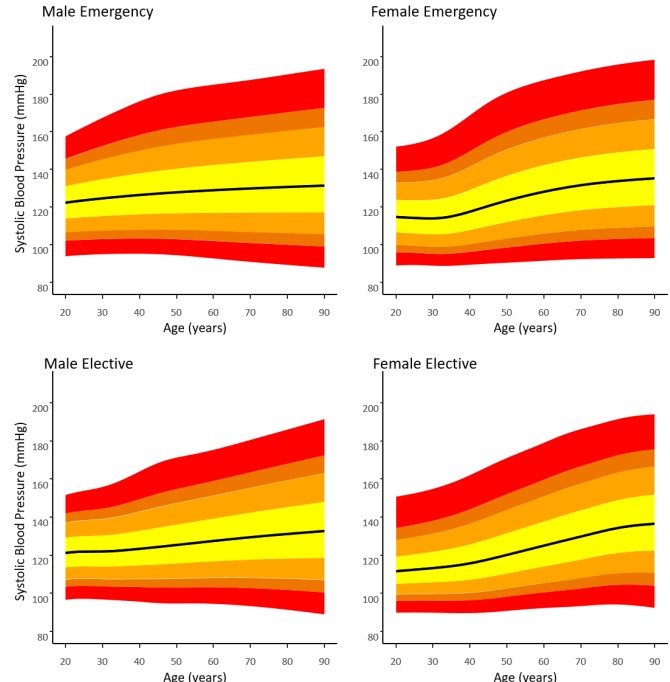

**Figure 4** 1st, 5th, 10th, 25th, 50th, 75th, 90th, 95th and 99th Systolic blood pressure centiles for emergency and elective subgroups.

**Figure 2** 1st, 5th, 10th, 25th, 50th, 75th, 90th, 95th and 99th centiles of (A) systolic, (B) diastolic and (C) pulse blood pressure for men and women between the ages of 20 and 90 years. Dashed lines in (A) denote SBP = (90, 100, 110) mm Hg.

cohort, the median DBP was 65 (CI: 64.6 to 65.0) mm Hg, 69 (CI: 69.0 to 69.4) mm Hg and 68 (CI: 67.6 to 68.2) mm Hg at ages 20, 50 and 90 years, respectively.

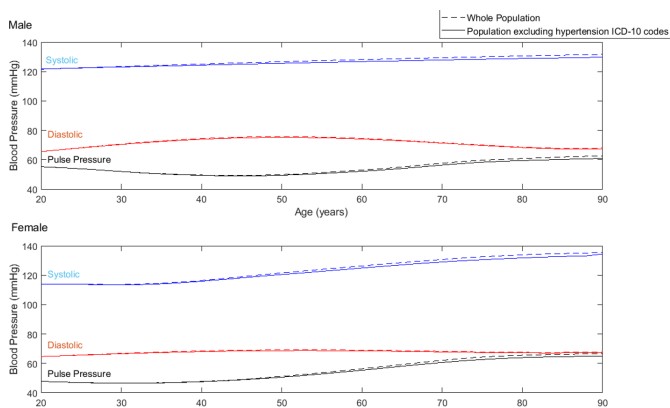

**Figure 3** Medians of systolic, diastolic and pulse blood pressure for all men and women between the ages of 20 and 90 (dashed lines) and the subgroup excluding patients with ICD codes for hypertension (solid lines).

For men, there was a modest reduction in median PP between the ages of 20 and 40 years from 55 mm Hg (CI: 54.6 to 55.9) to 50 mm Hg (CI: 49.2 to 50.0), whereas for women PP remained constant at 47 mm Hg (figure 2C). Median PP increases for both sexes between the ages of 40 and 90 years, from 50 mm Hg (CI: 49.2 to 50.0) to 63 mm Hg for men, and 48 mm Hg (CI: 47.6 to 48.0) to 66 mm Hg (CI: 65.8.6 to 67.2) for women. Bootstrapped CIs for SBP, DBP and PP are tabulated in online supplementary material appendix A. The online supplementary material also provides a post hoc analysis showing the centiles for the population that includes those that were readmitted within 7 days (online supplementary appendix B). There is no clinically meaningful difference between the posthoc analysis and the primary analysis.

Figure 3 shows the differences in medians for SBP, DBP and PP between the ages of 20 and 90 years for the whole study population in comparison to the subset without an ICD-10 diagnostic code for hypertension.

A total of 19 669 patients had an ICD-10 diagnostic code for hypertension. Of these, 24.0% (4711) had an SBP of <120 mm Hg and 2.3% (453) had a low SBP of <100 mm Hg at the time of discharge. Per-decade percentages were not calculated as small numbers of patients means that CIs are wider than any differences between decades.

Figure 4 shows SBP centiles for the emergency versus elective subpopulations. DBP and PP centiles are included in online supplementary appendix C. In the 24 hours prior to discharge, the 95th centile for SBP for emergency male admissions at 50 years was 163 mm Hg vs 155 mm Hg for elective male admissions. Similarly, the

95th centile for SBP for emergency female admissions at 50 years was 160 mm Hg vs 152 mm Hg for elective female admissions.

## Proportion of patients with blood pressure below published alert thresholds

Table 2 shows the cumulative percentages of men and women who had SBP less than 90, 100, and 110 mm Hg. These values correspond to the NEWS2 thresholds for hypotension.[7] 100 mm Hg is also a threshold used to assist in identifying sepsis.[8] For the 100 mm Hg threshold, 2.3% of younger (20–30 years) men and 3.5% of older men (81–90 years) fell below the threshold using their final reading in the 24 hours prior to discharge. In comparison, the criterion was met by 9.7% of younger women and 2.6% of older women.

## DISCUSSION

We have generated blood pressure centiles for age and sex from a large multi-hospital patient database.

Discharge blood pressures (SBP, DBP, PP) showed clinically significant differences across age ranges and by sex. SBP progressively increased with age for both sexes, but progression was greater in females. DBP increased and then decreased across the life course for both sexes. The fluctuation in DBP was greater for men than for women. These overall trends were visible in both the whole population, and for the cohort that did not have a diagnostic code for hypertension.

In populations outside hospital, these patterns are known to exist.[18–20] The Framingham studies showed that, for a healthy adult population, the mean arterial blood pressure increases throughout adulthood and that DBP decreases over the age of 60 years as SBP continues to rise.[21] Similar trends in both SBP and DBP have been shown in large cross-sectional cohorts from multiple countries.[22–24]

While the overall patterns for blood pressure are known to exist for the general population outside hospital, we believe that this is the first time that centiles have been derived from an in-hospital setting.

## Limitations

Vital signs were recorded as part of standard clinical practice, so the conditions for data recording were not controlled. This may have directly impacted the measured values. For instance, the state of wakefulness of the patient, which is known to be associated with blood pressure and pulse rate, was unknown.[25] However, it seems likely that such effects will be averaged out in a data set of this size.

We used the last recorded blood pressure in the 24 hours prior to discharge. While this lessens many biases in comparison to other methods and may represent the blood pressure recording when the patient is most stable, there may be other patterns at different points during a hospital admission.

Finally, this study uses a cross-sectional cohort, so the derived centiles may be affected by survival bias and birth cohort effects.[26 27] While the influence of these effects cannot be determined, we note that the overall trends follow those previously seen for longitudinal data in healthy populations.[28]

## Interpretation

The differences in normal vital sign ranges due to age and sex could have substantial implications for hospital practice. For example, table 2 showed that current SBP criteria for identifying sepsis (SBP <100 mm Hg) would be met by women much more frequently than by men up to age 50 year. Despite this, current evidence shows that men are more prone to develop sepsis than women.[29] A more accurate identification of patients at risk of sepsis may be possible through sex and age-stratified criteria.

A total of 19 669 patients diagnosed with hypertension had normal or low SBP immediately prior to discharge. This cohort may be reasonably assumed to be prescribed with anti-hypertensives for the purpose of managing blood pressure. While we do not have information on blood pressure medication following discharge, this

**Table 2** Percentages of male (N=36 185) and female (N=39 157) patients with low systolic blood pressure within each decade

| SBP | Gender (N,%) | Age (decade) | | | | | | | | |
|---|---|---|---|---|---|---|---|---|---|---|
| | | 18–20 | 21–30 | 31–40 | 41–50 | 51–60 | 61–70 | 71–80 | 81–90 | >90 |
| <90 | Male (120, 0.3%) | 0.2 | 0.3 | 0.1 | 0.2 | 0.2 | 0.3 | 0.5 | 0.5 | 0.8 |
| | Female (218, 0.6%) | 0.8 | 1.0 | 0.9 | 0.9 | 0.4 | 0.4 | 0.3 | 0.2 | 0.2 |
| <100 | Male (1063, 2.9%) | 2.9 | 2.3 | 2.4 | 2.2 | 2.6 | 3.2 | 3.5 | 3.5 | 4.6 |
| | Female (2060, 5.3%) | 11.1 | 9.7 | 9.4 | 6.5 | 4.4 | 3.1 | 2.6 | 2.6 | 2.0 |
| <110 | Male (4817, 13.3%) | 16.2 | 13.2 | 13.6 | 12.7 | 12.9 | 13.1 | 12.7 | 14.5 | 15.7 |
| | Female (8081, 20.6%) | 37.7 | 35.7 | 34.7 | 25.8 | 18.7 | 13.2 | 11.1 | 10.4 | 10.8 |

group may be considered an estimate of the upper-bound of those at risk of medication withdrawal. Inappropriate blood pressure medication withdrawal has been associated with higher risk of further complications.[30]

Another important application for age and sex stratification is Early Warning Scores (EWS). In these systems, integer scores are assigned to each vital sign according to deviation from normality. The aggregate score is then used to guide appropriate clinical care. One such EWS, the National Early Warning Score (NEWS), has been validated in a large in-hospital population and is widely used in the UK and the Ireland.[31]

Based on the results presented, an age-stratified score could strongly affect the quality of care a patient receives. For instance, from table 2, we see that 34.7% of women aged 31–40 years have a NEWS score of 1 or greater due to low SBP (SBP ≤110 mm Hg). In comparison, only 11.1% of women aged 71–80 years would meet the same blood pressure criterion. In contrast, 13.6% of men aged 31–40 years and 12.7% of men aged 71–80 years would have a NEWS score of 1 or more. These differences suggest it may be possible to improve discrimination between stability and deterioration by taking account of age and sex.

Until now, age and sex have not been included within any adult EWS, despite evidence indicating its limitations in predicting deterioration of elderly patients.[6 7] An update to the NEWS score to include these additional variables may be difficult to achieve in practice. In particular, the NEWS score was validated using in-hospital mortality. Adequate validation of the stratified score would require reasonable numbers of in-hospital mortality for each combination of sex and age-range, where death is relatively rare in younger cohorts. One alternative approach may be to derive a model that directly uses the representative centiles for each vital sign to provide EWS scores.[32]

### Generalisability

The data set was collected from all four hospitals, but we note that there are high proportions of white Caucasian patients. Previous studies have shown correlation between ethnicity and differences in blood pressure trajectory.[33] Whether inclusion of ethnicity could also improve discrimination requires further research.

Our work shows for the first time that meeting current blood pressure criteria for sepsis or early warning system alerts is substantially more likely in younger women than in all other groups in the 24 hours prior to discharge. Exploration of this finding is needed to determine whether adjustment for age and sex can improve discrimination and prevent overdiagnosis.

### CONCLUSION

Substantial variations in the final blood pressure recorded in the 24 hours prior to hospital discharge occur with age and sex. These result in large differences in the proportions of patients meeting the blood pressure criterion for sepsis and early warning score alerts. These factors should be examined to understand whether these factors could be used to improve discrimination between stability and deterioration.

**Contributors** DW, DAC and PJW conceived and designed the study; MAFP, DW and SG acquired the data; SG, DW and FS analysed the data. DW, SG, DAC, MAFP, FS and PJW were in involved in drafting and revising the manuscript and have approved the final submitted version.

**Funding** This publication presents independent research funded by the Health Innovation Challenge Fund (HICF-R9-524; WT-103703/Z/14/Z), a parallel funding partnership between the Department of Health and Wellcome Trust. The views expressed in this publication are those of the author(s) and not necessarily those of the Department of Health or Wellcome Trust. PJW and DAC are funded by the Health Innovation Challenge Fund and Wellcome Trust (HICF-R9-524 and WT-103703/Z/14/Z). PJW is supported by the NIHR Biomedical Research Centre, Oxford. SG is funded via an NIHR Doctoral Research Fellowship. FS is funded by the Rhodes Trust.

**Competing interests** DW and PJW co-developed the SEND e-Obs system (for which Sensyne Health has purchased sole license) from which the study database was collected. The company has a research agreement with the University of Oxford and royalty agreements with Oxford University Hospitals NHS Trust and the University of Oxford. DAC is Research Director of Sensyne Health. PJW is employed part-time and holds shares in Sensyne Health. DW undertakes consultancy for Sensyne Health.

**Patient and public involvement** Patients and/or the public were not involved in the design, or conduct, or reporting, or dissemination plans of this research.

**Patient consent for publication** Not required.

**Ethics approval** This paper is reported according to RECORD guidelines. Approval for obtaining the data used in this study was obtained from the Oxford Research Ethics Committee (REC ref: 16/SC/0264).

**Provenance and peer review** Not commissioned; externally peer reviewed.

**Data availability statement** Data are available upon reasonable request. The raw data for this research are not openly available. Researchers who present a sound analysis plan for any valid research can apply to ccrg@ndcn.ox.ac.uk. Proposals considered valid by the Kadoorie Critical Care Research Group Data Access Committee (which comprises independent researchers, clinicians, patient and public representatives) will be provided with data using the group's current system that complies with data governance requirements. R code and results in csv format are available at https://github.com/davcom2/BP_centiles.

**ORCID iDs**
David Wong http://orcid.org/0000-0001-8117-9193
Peter J Watkinson http://orcid.org/0000-0003-1023-3927

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
