## [Reviewer comments · BMJ Open]

ARTICLE DETAILS

TITLE (PROVISIONAL)	Cross-Sectional Centiles of Blood Pressure by Age and Sex: a four-hospital database retrospective observational analysis.
AUTHORS	Wong, David; Gerry, Stephen; Shamout, Farah; Clifton, David; Pimentel, Marco; Watkinson, Peter

VERSION 1 – REVIEW

REVIEWER	Pavel Hamet CHUM, University of Montreal, Canada
REVIEW RETURNED	20-Oct-2019

GENERAL COMMENTS	In this manuscript David Wong and colleagues evaluated sex and age dependence of blood pressure. These data have a potential for significant utility by its scope and size. Clinical utility here is focusing on risk of over-diagnosing sepsis, based on decline of blood pressure in hospital. Since the authors identified those diagnosed with hypertension, it would add much to utility of the study if an opposite question could be asked: is there a risk of antihypertensive medication withdrawal at the discharge in out of hospitalisation hypertensive subjects who become normo- or even low BP at the end of hospitalisation? From public health point of view, such a situation risks to be more prevalent than sepsis and this large study could contribute to our knowledge at this level. It is recognized the unwanted medication withdrawal represents a high risk of cardiovascular complications. See for instance withdrawal of medication at the end of intervention trial: Hirakawa at al. RISKS ASSOCIATED WITH PERMANENT DISCONTINUATION OF BLOOD PRESSURE LOWERING MEDICATIONS IN PATIENTS WITH TYPE 2 DIABETES J Hypertens;34:781-787, 2016
---

REVIEWER	Dr LF Gafane-Matemane North-West University, Hypertension in Africa Research Team South Africa
REVIEW RETURNED	25-Nov-2019

GENERAL COMMENTS	1. The abstract should mention the location/names of hospitals to give context.2. Did the authors ascertain that the conditions the patients were exposed to at the four different hospital settings were the same? If not, then this should be stated as a limitation.3. The inclusion of pulse pressure is not clear from the beginning of the manuscript as the measure is mentioned from the methodology section onward. PP should be briefly discussed in the introduction as well.
--

	4. The presentation of results can be improved, i.e. Figures 1 & 2 have no legends. Addition of percentages to the female and male columns will give a clearer picture of the findings in Table 1. The caption for Table 2 is in italics, and for Table 1 not. 5. There are some inconsistencies in the bibliography wherein the issue number is given for some references and not in others. Some letters are in caps or small letters while they should be. 6. Other technical errors are on page 3, line 29/30 & line 49/50, page 14, line 9
--	--

REVIEWER	Feng Dai Yale University
REVIEW RETURNED	23-Dec-2019

GENERAL COMMENTS	1. Page 12, line 14. As seen from table 2, it seems that the difference between two genders changed direction at age 60 (not 50). It would be helpful if authors could offer some explanations on the finding. Since patients in the table 2 were a mixture of patients with different admission characteristics, is it possible that this change of direction could be due to unadjusted confounding or other sources of bias? 2. Authors showed substantial variations in blood pressure by age stratification and sex. However, those do not necessarily mean that "meeting current criteria for sepsis or early warning system alerts is substantially more likely in younger women than in other groups" (Page 13, line 30). SBP is one criteria of SEPSIS and National warning score (NEWS) respectively. There is no guarantee that differences in one criteria will result in significant variations of overall score (i.e., SEPSIS, NEWS) itself by age and gender.
--

REVIEWER	Andrew Hayen Institution: University of Technology Sydney Country: Australia
REVIEW RETURNED	27-Dec-2019

GENERAL COMMENTS	 • The study rationale is not clear to me. For example, the authors state that "opportunities for earlier identification and management of patient deterioration may be being missed", but the population used is patients within 24 hours of discharge. Could the authors describe their rationale for the study further? Are these data meant to be useful for discharge planning – ie, is it safe to discharge a patient -- or for monitoring within a hospitalisation? • Why are patients who are readmitted within a week of discharge excluded? This introduces selection bias and limits the usefulness of the results -- how could a clinician know that a patient will be readmitted within 7 days of discharge? • Were any patients transferred from other (ie at admission) or to other hospitals at discharge? • I would like to see some description -- perhaps in a supplementary appendix -- of the actual models/distributions used for each vital sign. This is not made clear. • Please justify your model fitting method -- why is this appropriate? I am not sure that using different distributions for subgroups would lead to an unfair comparison. Does this mean the chosen distribution was fitted for each vital sign on all data? "To ensure fair comparison, the same distribution was chosen for all subgroups within any given vital sign."
---

	 • Could the authors make their R code available? It would also be useful if the centiles are made available in a format more easily read by a statistical package. • Could the authors explain what they mean in their data availability statement? "Any requests regarding data access should be made to ccrg@ndcn.ox.ac.uk". Under what conditions will the data be made available? • Is there a reason to give blood pressures to 1 and 2dp? • There are some spelling and grammatical errors in the manuscript. • What types of patients are the 40 patients who are not emergency or elective patients? • Table 1: I assume that >89 should be >=90.
--	--

VERSION 1 – AUTHOR RESPONSE

Reviewer: 1

In this manuscript David Wong and colleagues evaluated sex and age dependence of blood pressure. These data have a potential for significant utility by its scope and size. Clinical utility here is focusing on risk of over-diagnosing sepsis, based on decline of blood pressure in hospital. Since the authors identified those diagnosed with hypertension, it would add much to utility of the study if an opposite question could be asked: is there a risk of antihypertensive medication withdrawal at the discharge in out of hospitalisation hypertensive subjects who become normo- or even low BP at the end of hospitalisation? From public health point of view, such a situation risks to be more prevalent than sepsis and this large study could contribute to our knowledge at this level. It is recognized the unwanted medication withdrawal represents a high risk of cardiovascular complications. See for instance withdrawal of medication at the end of intervention trial: Hirakawa et al. RISKS ASSOCIATED WITH PERMANENT DISCONTINUATION OF BLOOD PRESSURE LOWERING MEDICATIONS IN PATIENTS WITH TYPE 2 DIABETES J Hypertens;34:781-787, 2016

Results:

19669 patients had an ICD-10 diagnostic code for hypertension. Of these, 24.0 % (4711) an SBP of <120 mmHg and 2.3% (453) had a low SBP of <100 mmHg at the time of discharge. Per-decade percentages were not calculated due as small numbers of patients means that confidence intervals are wider than any differences between decades.

Discussion:

19669 of patients diagnosed with hypertension had normal or low SBP immediately prior to discharge. This cohort may be reasonably assumed to be prescribed with anti-hypertensives for the purpose of managing blood pressure. Whilst we do not have information on blood pressure medication following discharge, this group may be considered an estimate of the upper-bound of those at risk of medication withdrawal. Inappropriate blood pressure medication withdrawal has been associated with higher risk of further complications [Hirakawa et al]

Reviewer: 2

Reviewer Name: Dr LF Gafane-Matemané

Institution and Country:

North-West University, Hypertension in Africa Research Team South Africa Please state any competing interests or state 'None declared': None declared

Please leave your comments for the authors below 1. The abstract should mention the location/names of hospitals to give context.

We are unable to provide any extra words, having reached the maximum word limit. We feel that any further detail would not be especially beneficial. For instance, highlighting that one of

the hospitals was the Horton hospital in Banbury would be largely meaningless for the vast majority of readers.

2. Did the authors ascertain that the conditions the patients were exposed to at the four different hospital settings were the same? If not, then this should be stated as a limitation.

The hospitals belonged to the same hospital trust, as mentioned in the text. This means that managerial oversight was the same at all sites.

3. The inclusion of pulse pressure is not clear from the beginning of the manuscript as the measure is mentioned from the methodology section onward. PP should be briefly discussed in the introduction as well.

We have now explicitly noted that our study objective includes centiles of systolic, diastolic, and pulse pressures.

4. The presentation of results can be improved, i.e. Figures 1 & 2 have no legends. Addition of percentages to the female and male columns will give a clearer picture of the findings in Table 1. The caption for Table 2 is in italics, and for Table 1 not.

- The legends for Figs 1 and 2 appear to have been dropped during the conversion to pdf. The captions should state:

Figure 1. Consort diagram showing analysis inclusion criteria

Figure 2. 1st, 5th, 10th, 25th, 50th, 75th, 90th, 95th and 99th centiles of systolic, diastolic and pulse blood pressure for males and females between the ages of 20 and 90. Dashed lines in (a) denote SBP = {90, 100, 110} mmHg

- Table 1 captions have been converted to italics

5. There are some inconsistencies in the bibliography wherein the issue number is given for some references and not in others. Some letters are in caps or small letters while they should be.

We have added issue numbers where possible. 'framingham heart' has been capitalized. Other titles have been kept the same. We note that National Early Warning Score is appropriately capitalized and is deliberately NOT capitalized in reference [30].

6. Other technical errors are on page 3, line 29/30 & line 49/50, page 14, line 9

We were unable to determine the errors here, and request further detail if amendments are required.

Reviewer: 3

Reviewer Name: Feng Dai

Institution and Country: Yale University Please state any competing interests or state 'None declared':
None declared

Please leave your comments for the authors below 1. Page 12, line 14. As seen from table 2, it seems that the difference between two genders changed direction at age 60 (not 50). It would be helpful if authors could offer some explanations on the finding. Since patients in the table 2 were a mixture of patients with different admission characteristics, is it possible that this change of direction could be due to unadjusted confounding or other sources of bias?

- **The differences did indeed change at 60. This has been amended in the text.**
- **Admission Main Specialty (Medical/Surgical/Other) was different for Men and Women and could be a possible source of confounding. However, subgroup analysis of centiles (unreported in manuscript initially, but now included in supplementary material) showed that there was minimal difference between Surgical and Medical specialties. In addition, these patterns have been observed in other large out of hospital cross-sectional populations. Given this, it is unlikely that the patterns are due to unadjusted confounding. Having said that, we acknowledge in the discussion that all results here are limited by interpreting cross-sectional data in a longitudinal way.**
- **As we are not blood pressure specialists, we are unable to provide a possible mechanistic explanation for the changes in DBP by age. However, we note that previous reports of blood pressure centiles have also not provided explanations, nor did we find any broad clinical agreement on the cause of the longitudinal changes during a literature search.**

2. Authors showed substantial variations in blood pressure by age stratification and sex. However, those do not necessarily mean that "meeting current criteria for sepsis or early warning system alerts is substantially more likely in younger women than in other groups" (Page 13, line 30). SBP is one criteria of SEPSIS and National warning score (NEWS) respectively. There is no guarantee that differences in one criteria will result in significant variations of overall score (i.e., SEPSIS, NEWS) itself by age and gender.

We agree fully and have clarified that the conclusion refers only to the blood pressure criterion of Sepsis and NEWS.

Reviewer: 4

Reviewer Name: Andrew Hayen

Institution and Country:

Institution: University of Technology Sydney

Country: Australia

Please state any competing interests or state 'None declared': None declared.

Please leave your comments for the authors below

- The study rationale is not clear to me. For example, the authors state that "opportunities for earlier identification and management of patient deterioration may be being missed", but the population used is patients within 24 hours of discharge. Could the authors describe their rationale for the study further? Are these data meant to be useful for discharge planning – ie, is it safe to discharge a patient -- or for monitoring within a hospitalisation?

These centiles are descriptive and intended to convey information about the stable hospital population (excluding those who are unstable including death or readmission). We assume that this patient group become more stable as hospital stay progresses, such that the blood pressures prior to discharge represents the most 'normal' value.

The stable group may be used to infer unstable patients (whether general deterioration, or specific deterioration such as sepsis) via one class classification/ novelty detection, which has previously been used when the minority class is relatively uncommon (e.g. Centile Early Warning scores from Tarassenko et al.)

- Why are patients who are readmitted within a week of discharge excluded? This introduces selection bias and limits the usefulness of the results -- how could a clinician know that a patient will be readmitted within 7 days of discharge?

The focus of this work was to present the stable hospital population. We considered those who were readmitted in a short span as those likely to be in an unstable condition (but potentially incorrectly discharged). Of course, a clinician would not know whether a patient were to be readmitted at the time of discharge. However, we have not attempted to predict an outcome conditioned on readmission here.

- Were any patients transferred from other (ie at admission) or to other hospitals at discharge?

A small proportion of patients were transferred from other hospitals (i.e. not from the four hospitals included here), according to the coded data. Given that the main teaching hospital in our study caters for multiple specialties, it is likely that many transfers IN were for patients requiring higher levels of care.

A similar proportion were transferred OUT to other hospitals. These are likely to be primarily stabilized patients, for the same reason as above. We provide the coded values below for interest, but note that the broad code groupings mean that nuance concerning the acuity of patients will be missed:

49 NHS other Hospital Provider - high security psychiatric accommodation in an NHS Hospital Provider (NHS Trust or NHS Foundation Trust) 35

- 51 NHS other Hospital Provider - WARD for general PATIENTS or the younger physically disabled or A & E department 2875
- 52 NHS other Hospital Provider - WARD for maternity PATIENTS or Neonates 15
- 53 NHS other Hospital Provider - WARD for PATIENTS who are mentally ill or have Learning Disabilities 33

Similarly for discharge destination:

- 49 NHS other Hospital Provider - high security psychiatric accommodation **5**
- 50 NHS other Hospital Provider - medium secure unit **10**
- 51 NHS other Hospital Provider - WARD for general PATIENTS or the younger physically disabled **3270**
- 52 NHS other Hospital Provider - WARD for maternity PATIENTS or Neonates **2**
- 53 NHS other Hospital Provider - WARD for PATIENTS who are mentally ill or have Learning Disabilities **115**

- I would like to see some description -- perhaps in a supplementary appendix -- of the actual models/distributions used for each vital sign. This is not made clear.

A variety of distributions were assessed to find the best fit for each vital sign. The Box-Cox t distribution was the best fit for male and female SBP, the Skew t type 3 distribution was chosen for male DBP and male and female PP, and the Skew power exponential distribution was chosen for female DBP. SBP and PP models used penalised-splines as a smoother, whilst the DBP models used fractional polynomials as a smoother.

- Please justify your model fitting method -- why is this appropriate? I am not sure that using different distributions for subgroups would lead to an unfair comparison. Does this mean the chosen distribution was fitted for each vital sign on all data? "To ensure fair comparison, the same distribution was chosen for all subgroups within any given vital sign."

We found that using different distributions and smoothing functions can result in slightly different appearances in the centile plots. For example, using fractional polynomial smoothers tended to result in slightly smoother centiles than penalized splines. The main reason for looking at centiles within subgroups was to compare between groups. Therefore we did not want to falsely create the impression that there were differences by using different distributions or smoothing functions. All models were well fitted to the data.

- Could the authors make their R code available? It would also be useful if the centiles are made available in a format more easily read by a statistical package.

The R code is now available via github (https://github.com/davcom2/BP_centiles). Whilst BMJ Open only accepts pdf supplementary files, we have also included a csv of the centiles within the github repository.

- Could the authors explain what they mean in their data availability statement? "Any requests regarding data access should be made to ccrg@ndcn.ox.ac.uk<<mailto:ccrg@ndcn.ox.ac.uk>>". Under what conditions will the data be made available?

We have updated the data availability statement to provide clearer guidance on conditions for data access:

Researchers who present a sound analysis plan for any valid research will apply by ccrg@ndcn.ox.ac.uk. Proposals considered valid by the Kadoorie Critical Care Research Group Data Access Committee (which comprises independent researchers, clinicians, patient and public representatives). Data will be provided using the group's current compliant system.

- Is there a reason to give blood pressures to 1 and 2dp?

BP is reported to the nearest integer throughout the manuscript. Confidence intervals are expressed to 1 d.p., reflecting uncertainty in the centile estimate. Reporting CIs to the nearest integer would give a false impression of minimal uncertainty, as CI intervals were typically smaller than 1 mmHg

- There are some spelling and grammatical errors in the manuscript.

We have corrected two typos. Please advise on any further grammatical errors if they require correction.

- What types of patients are the 40 patients who are not emergency or elective patients?

As we state in the text, the codes are defined in full within the NHS dictionary. 'Other' admissions include ante- and post-partum admissions and transfers from other hospital providers for non-emergency care.

- Table 1: I assume that >89 should be >=90.

We have amended the relevant row label

VERSION 2 – REVIEW

REVIEWER	Pavel Hamet Université de Montreal, Canada, P.Q.
REVIEW RETURNED	15-Feb-2020

GENERAL COMMENTS	Authors have responded fully satisfactory to mine previous comments and discussed issuing limitations. Paper is important and useful , should be made available to clinicians.
--

REVIEWER	Dr Lebo Gafane-Matemane North-West University, Physiology, Hypertension in Africa Research Team
REVIEW RETURNED	18-Feb-2020

GENERAL COMMENTS	Below are errors that were not corrected: Abstract (spacing): (CI: 75.5-75.9)mmHg Competing interests (remove) : and.
---

REVIEWER	Feng Dai Yale University
REVIEW RETURNED	27-Feb-2020

GENERAL COMMENTS	I appreciate authors' efforts in revising the manuscript. I have no more comments related to the manuscript.
--

REVIEWER	Andrew Hayen University of Technology Sydney
REVIEW RETURNED	11-Mar-2020

GENERAL COMMENTS	I remain unclear why the authors have included patients who are subsequently readmitted. While I understand that the authors wish to have a set of stable patients, there is bias induced by removing this group. Otherwise, the authors have addressed my comments.
---

VERSION 2 – AUTHOR RESPONSE

David Wong (on behalf of the authors)

Reviewer: 2

Please leave your comments for the authors below Below are errors that were not corrected:

Abstract (spacing): (CI: 75.5-75.9)mmHg Competing interests (remove) : and.

** Thank you – we have now corrected these errors.**

Reviewer: 4

Reviewer Name: Andrew Hayen

Please leave your comments for the authors below

I remain unclear why the authors have included patients who are subsequently readmitted. While I understand that the authors wish to have a set of stable patients, there is bias induced by removing this group. Otherwise, the authors have addressed my comments.

** We have included additional supplementary analysis including the patients who required readmission. There is minimal difference to the original analysis (<1mmHg for each centile at all ages)**